psychology

overimitation, tool functionality, previous success, social learning

**Author for correspondence:**
Aurélien Frick
e-mails: aurelien.frick@ed.ac.uk, aurelien.frick@hotmail.com

† Shared first authorship.

# Carry-over effects of tool functionality and previous unsuccessfulness increase overimitation in children

Aurélien Frick[1,2,†], Hanna Schleihauf[3,4,†],
Liam P. Satchell[5] and Thibaud Gruber[6]

[1]Department of Psychology, University of Edinburgh, Edinburgh, UK
[2]Laboratory of Experimental Psychology, Suor Orsola Benincasa University, Naples, Italy
[3]Department of Psychology, University of California, Berkeley, CA, USA
[4]Cognitive Ethology Laboratory, German Primate Center, Göttingen, Germany
[5]Department of Psychology, University of Winchester, Winchester, UK
[6]Faculty of Psychology and Educational Sciences and Swiss Center for Affective Sciences, University of Geneva, Geneva, Switzerland

AF, 0000-0002-3702-5186; HS, 0000-0003-4344-9070;
TG, 0000-0002-6766-3947

Children 'overimitate' causally irrelevant actions in experiments where both irrelevant and relevant actions involve a single common tool. This study design may make it harder for children to recognize the irrelevant actions, as the perceived functionality of the tool during the demonstration of the relevant action may be carried over to the irrelevant action, potentially increasing overimitation. Moreover, little is known how overimitation is affected by the demonstrator's expressed emotions and the child's prior success with the task. Here, 131 nine- to ten-year-old French and German children first engaged in a tool-based task, being successful or unsuccessful, and then watched an adult demonstrating the solution involving one irrelevant and one relevant action before smiling or remaining neutral. These actions were performed with the same tool or with two separate tools, testing potential carry-over effects of the functionality of the relevant action on the irrelevant action. We show that overimitation was higher when the same tool was used for both actions and when children were previously unsuccessful, but was not affected by the demonstrator's displayed emotion. Our results suggest that future overimitation research should account for the number of tools used in a demonstration and participants' previous task experience.

# 1. Introduction

Tools—defined as external objects detached from their substrate used to attain a goal [1]—are an essential part of our lives. As humans, we use a large variety of tools ranging from simple (e.g. a wooden stick to stir) to more technologically sophisticated ones (e.g. a laptop to chat with a friend). However, other animals have also been found to use tools. For instance, bottlenose dolphins (*Tursiops aduncus*) use sponges [2], and chimpanzees (*Pan troglodytes*), the most prolific non-human tool-users, use a variety of tools such as sticks, sponges or hammers (e.g. [3,4]). Note that tool-use or tool-making is not limited to mammals, as bird species, particularly corvids, such as New Caledonian crows (*Corvus moneduloides*), manufacture and use tools in a variety of contexts (e.g. [5–8]). Nevertheless, the technological advance of human tool use is unique in the animal kingdom. It has been proposed that the complex tool advances of humankind are the results of their unique abilities for accumulating modified knowledge from previous generations generating more sophisticated repertoires over time, a process known as cumulative culture ([9–11]; but see [12,13]).

Imitation, alongside with innovation, is thought to play a key role in the process of cumulative culture ([14]; but see [15]). Children are particularly skilful at acquiring tool-using behaviours through imitation. Interestingly, they tend to reproduce both causally relevant and irrelevant actions to retrieve a reward [16]. This behaviour termed 'overimitation' [17] has been the focus of more than a decade of intensive research (for reviews see [18,19]). A recent definition for this phenomenon is the 'imitation of perceivably causally unnecessary actions in relation to the goal of an action sequence performed by a model' [18, p. 91]. The few comparative studies that have been conducted suggest that this phenomenon is absent in non-human primates [16,20,21], indicating its potential uniqueness in the human lineage (but see [22,23] for possible evidence in dogs). Developmental research in humans has revealed that there are early signs of overimitation at around 18 months [24], becoming more evident from 2 years old on [25], and gradually increasing from 3 to 5 years old, before reaching higher fidelity to the actions demonstrated in adulthood [26–28]. Moreover, cross-cultural comparisons point out that overimitation also occurs in non-WEIRD (Western, Educated, Industrialized, Rich and Democratic; [29]) cultures, with some degrees of variation as compared with WEIRD cultures, such as manifesting later in the development and being less frequent in some human populations [30–35].

The theories proposed to explain overimitation can be parsed into two groups. The first group sees overimitation as a result of erroneous reasoning, meaning that participants fail to distinguish causally relevant from causally irrelevant actions, and falsely encode others' intentional actions as relevant to the goal of the task and therefore reproduce them [17,36]. Conversely, the second group of theories converges on the view that the imitator recognizes that the superfluous action is not relevant to reaching the instrumental goal of the action sequence, but nevertheless reproduces the action for social, such as affiliative [37] or normative purposes [38]. We acknowledge that the affiliative and normative accounts are different from each other. The affiliative account focuses essentially on the individual affiliation between the observer and the demonstrator, the normative account focuses on the propensity to belong to a social group and therefore follow the social norms of this group. Here, we will consider that both the affiliative and normative accounts fall under the umbrella of social motivations. While the automatic causal hypothesis predicts a systematic emergence of overimitative behaviour whenever an intentionally performed action is observed, the social motivation hypothesis argues for a more flexible occurrence of overimitation depending on the social context.

By now many studies have explored the social motivations of overimitation. For example, studies have shown that the presence of the model who performed the irrelevant actions increases overimitation [33,37], and there is increased overimitation after observing live-performed demonstrations as opposed to video-performed demonstrations [27]. Social encouraging situations and playful contexts increase children's overimitation tendencies [24,39,40]. Overimitation rates are especially high when tasks are framed as normative [38,41,42] or after children are primed with third-party ostracism [43]. Moreover, the demonstrator's sex seems to have an influence: boys copy actions performed by a male or female model equally, while girls copy irrelevant actions less when demonstrated by a male model [44]. These findings highlight that children's overimitation tendencies are sensitive to the different social contextual factors. Taken together, these results support the idea that overimitation results from various underlying social motivations.

Nevertheless, social explanations fail to explain why children still overimitate when they are left alone after the experimental session is over [17] or even in naturalistic situations without any social factors that could explain the persistence of overimitation [45]. Moreover, only erroneous reasoning explanations could support the fact that children more frequently overimitate irrelevant actions that involve

physical contact with the object containing a reward compared with irrelevant actions which involve no such contact [35,46,47]. Not many studies have yet investigated which factors could increase children's likelihood to make false conclusions about the causal necessity of the demonstrated actions (but see [17,36,48]).

The current study aims to investigate two factors which might increase children's tendency to falsely classify irrelevant actions as being relevant, and one factor which might increase children's social motivations to imitate.

The first factor which might influence children's erroneous causal reasoning is whether irrelevant and relevant actions are demonstrated with an identical tool or different tools. A recent finding suggests that not all irrelevant actions, even if they involve physical contact with the reward containing object, are copied to the same extent. Frick *et al.* [48] observed lower rates of overimitation when the irrelevant action was performed with a different tool (i.e. circling a bottle containing a reward with a string) than the relevant action (i.e. making a hook with a pipe-cleaner). Indeed, it is notable that tool-based overimitation studies reporting higher rates of overimitation have often relied upon the same tool to perform both relevant and irrelevant actions (see [18]), although some other research on overimitation has used non-tool-based tasks, such as necklace making [49,50]. This procedure might have led to carry-over effects related to the perceived functionality of the tool involved in both the irrelevant and relevant actions. Demonstrating different actions with different tools might in contrast encourage separate evaluations of the functionality of each action. As such, performing irrelevant and relevant actions with the same tool could increase overimitation while performing irrelevant and relevant actions with different tools could reduce overimitation. Answering this question has the potential to shed new light on our understanding of tool-related overimitation and factors increasing the risk of erroneous causal reasoning.

Another factor which might lead to more mistakes in children's causal reasoning, is whether they have some prior knowledge about this task. If children have an understanding about the functionality of the task, they might be less likely to copy irrelevant action, whereas they might be more likely to do so if they have not figured out beforehand how the task can be solved. In most overimitation studies children are directly presented with a demonstration of irrelevant and relevant actions with little or most often no chance to explore the apparatus beforehand (e.g. [16,20,27,28,30,32,34,51]); but see [18]). Only a few studies gave children the chance to get some prior experience with the apparatus or some insights into its functionality (e.g. [47,52]). Of particular interest, Wood *et al.* [52] investigated whether children's overimitation rates changed when they had or had not a chance to acquire some information about the functionality of the apparatus, before the action sequence with irrelevant and relevant actions was demonstrated. In this study, children who acquired a solution to the task on their own overimitated less than children with no such experience. However, to our knowledge, there is no study that focuses on the influence of children's performance during an exploration phase (i.e. whether children successfully or unsuccessfully retrieved a reward). As such, it is still unknown how a prior successful or unsuccessful experience in a task influences the subsequent tendency to overimitate. In everyday life, it is quite common that one receives demonstrations after an unsuccessful personal attempt or a demonstration of an alternative and maybe 'more correct' way to do something even if one was successful before. Thus, looking at the effects of prior success on children's tendency to overimitate will give us new insights in how task understanding and familiarity influence the phenomenon.

One factor which might influence children's social motivations to overimitate that has so far been neglected in overimitation research is the influence of a demonstrator's expressed emotions. Interestingly, placing children in a playful context seems to increase their tendency to reproduce irrelevant actions ([40,47]; but see [18]). Since social interaction and pedagogical cues do not influence overimitation *per se* [46], the effect of the playful context may be linked to its induction of an emotional bond between child and model. For instance, the subtle emotional facial expression of an interaction partner strongly influences object preference with objects being more liked when the face looking at them smiles slightly [53,54]. As such, subtle emotional cues might potentially influence the transmission of culturally relevant knowledge such as tool-use skills via social learning mechanisms, such as overimitation. However, even though emotions are discussed to be playing a key role in social learning [55], to date no study has investigated the potential role of the demonstrator's portrayed emotions on overimitation.

In sum, the present study sets to examine the potential influence of three factors: the number of tools used for the demonstration of irrelevant and relevant actions, a successful versus an unsuccessful experience with the task prior to the demonstration, and the emotion displayed by the demonstrator. To this aim, we presented German and French 9- to 10-year-old children with a tool-based problem, the Hook task [48,56]. First, they got a chance to solve the task on their own. Then, they watched a demonstration of an irrelevant and a relevant action, performed with either one tool or two tools by

a demonstrator who either remained neutral or smiled at the end of the demonstration. We targeted 9- to 10-year-old children, since it has been found that at that age approximately 50% of the children can successfully solve the task [48,56].

We predicted higher rates of overimitation when irrelevant and relevant actions were performed with the same tool, when children were previously unsuccessful in solving the task on their own, and when the emotion displayed by the model was positive.

# 2. Methods

## 2.1. Participants

Participants were 131 French ($n = 63$) and German ($n = 68$) 9- to 10-year-old children ($M_{age} = 9.87$ years, s.d.$_{age} = 0.52$ years, range = 9.02–10.97 years, 58 girls). This sample size was achieved by the maximum number of children available in the participating schools. Based on the location of the schools where the data was collected, we estimate that most children came from middle to high socio-economic backgrounds (SES) and were mostly Caucasian (92% of the children who took part in this study based on the video recordings). Parental consent was obtained for each child and all children received stickers during participation. This study received approvals from the Ethics Committee of the University of Edinburgh, Edinburgh, Scotland and Max Planck Institute of Evolutionary Anthropology, Leipzig, Germany, as well as from the participating schools.

## 2.2. Materials and procedure

Children were tested in a quiet room in isolation from other children within their schools. A male French native speaker (A.F.) and a female German native speaker (H.S.) conducted the experimental sessions in France and Germany, respectively. The instructions were initially written in English and then translated into French and German by A.F. and H.S. to ensure reliability across locations. Each experimental session consisted of (i) a pre-demonstration phase, followed by (ii) a demonstration phase and (iii) a post-demonstration phase.

(i) In the pre-demonstration phase, children were presented with the Hook task consisting of a transparent glass bottle attached to a wooden base containing a plastic bucket with an animal sticker inside. The experimenter informed the children that their task was to retrieve the animal sticker from the bottle without putting the bottle upside-down and that if they succeeded in doing so, they would be allowed to keep the sticker. Importantly, the opening of the bottle was too small for the children to reach the bucket or the reward with their hands. Subsequently, the experimenter put a black or white straight pipe-cleaner and a black or white piece of string next to the bottle (colour and side counter-balanced) saying 'Here are two things that might help you to retrieve the sticker from the bottle'. Then, children were given 1 min to get the reward to ensure comparison with previous studies [48,56]. Children who successfully retrieved the sticker from the bottle within 1 min were categorized as being successful. Children who failed to get the reward within the 1 min limit were categorized as being unsuccessful.

(ii) In the demonstration phase, children were shown a short video clip in which a non-familiar male or a female demonstrator (coming from a different locality than where the children were tested) performed an irrelevant followed by a relevant action in order to retrieve the reward. The presentation of the male or female demonstrator was counter-balanced between boy and girl children. Children were randomly assigned to the one-tool condition or the two-tools condition, as well as to the emotion or no-emotion condition. In the one-tool condition, the model performed irrelevant and relevant actions with the same tool. The model first made a loop at the top end of the pipe-cleaner (irrelevant action) before forming a hook with the bottom end (relevant action). In the two-tools condition, the models used different tools for the irrelevant and relevant action. They first used a string to form a circle around the bottom of the bottle (irrelevant action) and then formed a hook with one end of the pipe-cleaner (relevant action). Importantly, the demonstration did not include the retrieval of the reward. Thus, children observed the relevant making of the hook, but did not see the hook in action. In the emotion condition, the model smiled after performing the two actions, whereas in the no-emotion condition the model remained neutral.

(iii) In the post-demonstration phase, children were again presented with the Hook task and the experimenter announced that they had another chance to solve the Hook task: 'Now can you try again to retrieve the reward out of the bottle?' and children were given another minute to do so. During the complete procedure, the experimenter remained present in the room.

## 2.3. Coding

For the pre-demonstration phase, we coded whether or not children successfully retrieved the reward from the bottle (e.g. by making a hook, dragging the bucket with the pipe-cleaner out of the bottle), categorized as successful or unsuccessful, respectively. For the post-demonstration phase, children assigned to the one-tool condition were considered as overimitators if they made a loop with the pipe-cleaner regardless of whether this loop was closed and whether this irrelevant action preceded the relevant action or not. Children assigned to the two-tools condition were considered as overimitators if they circled the bottle with the string regardless of the precision with which they reproduced the irrelevant action (e.g. whether the circle faced them or not as in the demonstration). A.F. coded all the videos, and a second coder blind to the hypotheses coded around 50% of the videos ($N = 66$) with $\kappa = 0.97$ indicating a near-perfect level of agreement for coding overimitative behaviours.

## 2.4. Data analyses

We analysed the binary outcome variables (success versus no success, overimitation versus no overimitation) using a logistic generalized linear model (GLM) in R [57] using the *glm* function from the *lme4* package [58]. For the pre-demonstration phase, we fitted a GLM with success in the pre-phase as a response variable and only the intercept as the predictor. We used a Wald test to compare whether the numbers of successes differed from chance level. To see whether success in the pre-demonstration phase influenced success in the post-demonstration phase, we fitted a GLM with the response of post-demonstration success and the predictor pre-demonstration success. Next, we conducted a pre-analysis to see whether the effect of children's or demonstrator's sex had an effect on children's overimitation. Since children's sex turned out to influence children's overimitation tendencies ($\chi^2 = 6.479$, d.f. = 1, $p = 0.011$, Nagelkerke's $R^2 = 0.065$), we included this predictor in the main analysis. Demonstrator's sex had no effect on children's overimitation ($\chi^2 = 0.158$, d.f. = 1, $p = 0.691$, Nagelkerke's $R^2 = 0.002$). For the main analysis, we then generated a full model with all predictor variables of interest and all possible two-way interactions. The interactions were included exploratorily and we did not expect any interaction effects. We followed the recommendation to have between 5 and 10 events per variable (EPV) calculated on the basis of the number of positive outcomes divided by the number of predictors included in the model (see [59,60]). Indeed, a total of 74 children overimitated (positive outcome) and our full model of interest contained 11 predictors, as such our EPV was of 6.72. In sum, we examined whether the *number of tools* (one versus two), *previous experience* (success versus no success), *emotion of the demonstrator* (positive versus neutral) and *sex of the child* (boy versus girl), and possible two-way interactions of these variables had an effect on the probability that children overimitated. The assumption of the absence of over-dispersion and the absence of collinearity were fulfilled and the model proved to be stable (details are reported in the electronic supplementary material). To avoid an increased type 1 error risk due to multiple testing, we first tested the overall effect of the test predictors. Therefore, we compared the full model's deviance with that of a null model comprising only the intercept to examine whether the inclusion of the fixed effects provided a better fit to the data. When comparison to the null model proved to be significant, we could test the effect of the single predictors. To do so we compared the full model with the corresponding reduced models that lacked the predictor of interest. Since none of the interactions turned out to be significant (details are reported in the electronic supplementary material) we fitted a new model with only the main effects and repeated the procedure. Confidence intervals were derived using the function *confint* of the package *stats*.

# 3. Results

## 3.1. Pre-demonstration phase

Of the children, 69% were successful in retrieving the reward in the Hook task, while 31% remained unsuccessful. The Wald test indicated that this difference was significant $p < 0.001$. All successful children made a hook to retrieve the reward except one child who dragged the bucket with the pipe-cleaner to retrieve it.

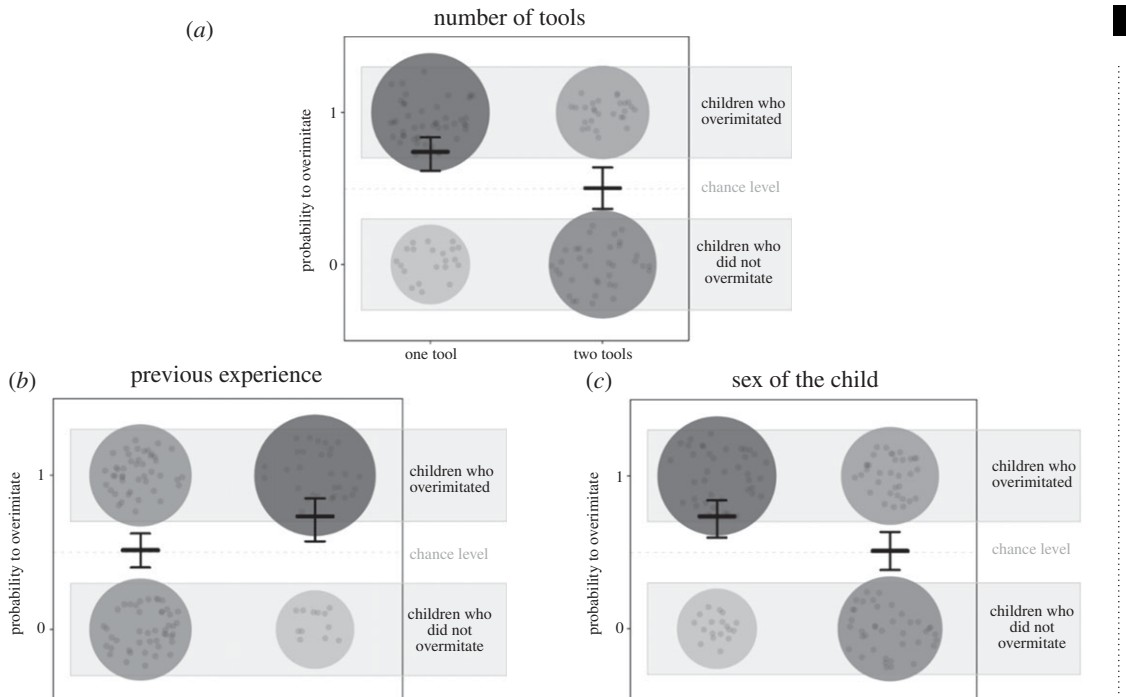

**Figure 1.** Probability of children showing overimitation as a function of *number of tools* (*a*), *previous experience* (*b*) and *sex of the child* (*c*). The number of children who overimitated (1) or who did not overimitate (0) is represented by the size and darkness of the large circles (bigger size and darkness represent larger number of overimitators) as well as by the number of small circles (each small circle represents one child). Lines represent the point estimates for the main effects of the GLM (centred for the factors not depicted) with the corresponding 95% confidence intervals that were calculated with parametric bootstraps for all trial analyses and with the function *confint* of the package *stats* for the first trial analysis.

## 3.2. Post-demonstration

Of the children, 96.2% were successful in retrieving the reward in the post-demonstration phase within 1 min (all making a hook to retrieve the reward). Children's success in the pre-demonstration phase did not influence their success in the post-demonstration phase ($\chi^2 = 0.334$, d.f. = 1, $p = 0.563$).

The full model provided a significantly better fit to the data than the null model ($\chi^2 = 24.542$, d.f. = 130, $p < 0.006$, Nagelkerke's $R^2 = 0.230$). Our model revealed no significant interaction effects (see the electronic supplementary material), but significant main effects for the factors *number of tools* ($\chi^2 = 7.66$, d.f. = 1, $p = 0.006$, Nagelkerke's $R^2 = 0.079$), *previous experience* ($\chi^2 = 5.25$, d.f. = 1, $p = 0.022$, Nagelkerke's $R^2 = 0.055$) and *sex of the child* ($\chi^2 = 6.57$, d.f. = 1, $p = 0.010$, Nagelkerke's $R^2 = 0.068$). The factor *emotion of the demonstrator* was not significant ($\chi^2 = 0.34$, d.f. = 1, $p = 0.563$, Nagelkerke's $R^2 = 0.003$). More specifically, our model predicts a higher probability for children to overimitate in the one-tool condition (74% probability to overimitate) than in the two-tools condition (50% probability to overimitate); a higher probability for children to overimitate when they were previously unsuccessful (73% probability to overimitate) compared with successful (51% probability to overimitate); and finally, that girls are more likely to overimitate (74% probability to overimitate) than boys (51% probability to overimitate). Note that for the factor *emotion of the demonstrator*, the model predicted that participants were not significantly likely to overimitate more if the demonstrator was smiling (65% probability to overimitate) versus a demonstrator with a neutral facial expression (60% probability to overimitate). Model estimates and confidence intervals are reported in figure 1 and electronic supplementary material, table S5.

## 4. Discussion

As expected, using the same tool in the demonstration of the irrelevant and the relevant action led to more overimitation compared with when the irrelevant action was performed with a different tool than the relevant action. This result speaks for carry-over effects related to the perceived functionality of the tool

when the same tool is involved in both types of actions. The evaluation of the tool as being relevant seems to carry over the appearance of relevance to the non-functional action involving the same tool. By contrast, when different tools were involved in the demonstration of the relevant and irrelevant action, children appeared to evaluate the functionality of each action separately. Alternatively, this result can be explained by the different types of irrelevant actions that are circling the bottle with the piece of string and making a loop with the pipe-cleaner. Distinguishing between irrelevant actions, Schleihauf & Hoehl [61] recently argued that pseudo-instrumental actions (actions that are similar to instrumental actions) are more likely to be overimitated than non-contact actions and superficial contact actions. In our case, the irrelevant action presented in the two-tools condition was a superficial contact action, as the piece of string touched the reward container, while the irrelevant action presented in the one-tool condition could be classified as a pseudo-instrumental action, as this action was very similar to the relevant action. Consequently, children might have overimitated more when one tool was used than when two tools were used due to the difference in the type of irrelevant actions. A limitation of the irrelevant action used here in the one-tool condition is that the loop may have been thought to be a handle by the children, and therefore perceived as perhaps not necessary, but not totally irrelevant. As such, future studies disentangling different types of irrelevant actions are needed to confirm our results. All in all, this result offers an interesting avenue of research for future tool-based overimitation studies to examine the potential confound of the number of tools and the type of irrelevant actions. Such evaluation is crucial to enhance our understanding of why children are more likely to overimitate in a given tool-based situation than in another.

Consistent with our hypothesis, previously unsuccessful children overimitated more than previously successful children. This result is in line with Wood *et al*. [52], in which they found that children who gained personal experience on how to solve a problem were less likely to imitate irrelevant actions afterwards compared with children who were given socially provided information of how to solve the problem. However, other previous results point in a different direction. Frick *et al*. [48] observed low rates of overimitation in unsuccessful children. But these authors always used two tools in their demonstration whereas here, children also watched a demonstration involving only one tool, thus enhancing overimitation rates. More specifically, the irrelevant action was the only new information for successful children, whereas both the relevant and irrelevant actions were new for unsuccessful children, giving different weights to these actions [62]. In the present study, successful children did not take the newly performed irrelevant action into account, and favoured the already known relevant action to retrieve the reward. Conversely, unsuccessful children might have considered both actions as relevant for the instrumental goal of retrieving the reward. Interestingly, this result can be an important venue for future comparative research. Only three studies have been carried out in non-human primates [16,20,21], reporting no overimitative behaviours. However, these studies have all placed apes in a situation in which they first saw the irrelevant and relevant actions performed by a human demonstrator before interacting with the puzzle box. Further investigations of overimitation occurrence in non-human animals in which they first perform an ecological problem-solving task, allowing, therefore, to classify them as successful or unsuccessful, before seeing a demonstration of relevant and irrelevant actions performed by a conspecific, has the potential to enhance our understanding of overimitation in these species, with the prediction that unsuccessful individuals might present overimitative behaviours (for a study doing this on non-human primates imitation, see [10]).

By contrast to our prediction, we found similar rates of overimitation regardless of the emotion displayed by the model, suggesting that positive emotion does not facilitate overimitation. This is surprising given that social learning appears to depend in part on emotions, and emotions favour the transmission of goal-directed behaviours [55,63,64]. However, it is possible that our design failed to model a strong positive emotion as the model was only smiling at the end of the demonstration. Another way to further test the effect of emotion would be to have a demonstration in which the model constantly probes children with positive or playful sentences when manipulating the tools such as 'Look! This is very funny what I am doing' versus more neutral sentences such 'Look! This is what I am doing' as previously done in other studies (e.g. [65]). Similarly, emotional self-reports by the participants could offer a venue to investigate the role of emotions. Furthermore, it is also possible that neutral and positive emotions are equally motivating, whereas negative emotions could inhibit overimitation. As such, further studies are needed to more thoroughly test the influence of emotions in context.

As reported in some previous studies [44,48,66,67], we found a sex effect during overimitation, which confirms that sex differences are important individual-level factors to consider for overimitation. More specifically, girls overimitated more than boys. While this result supports the hypothesis that girls conform more than boys to a demonstration [66,68] and is in line with

comparative data in chimpanzees [69], which indicate a general trend in the hominine lineage for females to be more inclined to acquire tool-use skills through social learning, it goes against the findings from Frick *et al.* [48] in the same task showing that boys overimitated more than girls. One plausible explanation for the observed difference is that it results from different experimental settings. First, we used video clips rather than live performances in this experiment, although we are not aware of sex effects linked to the way an experiment is administered. Another interesting possibility is that studies reporting more overimitation in boys than girls have essentially focused on children around 5 years of age. Although Frick *et al.* [48] did test 5- to 12-year-olds, only unsuccessful children in the Hook task were tested, drastically reducing the number of children older than 9 years old. As such, there might be some age-related differences regarding the sex effect during overimitation, with boys overimitating more than girls at a young age, but with this pattern reversing at a later age when using the Hook task in the type of demonstration used in Frick *et al.* [48] and here. Future studies should more thoroughly test this assumption as the age range of the children tested in the present study cannot allow examining the interaction between age, sex and number of tools on overimitation, nor whether this relates more to the causal or social dimensions of overimitation. Another interesting avenue for further investigations concerns how the effects of the number of tools and the sex of the child are subjected to age-related differences. Perhaps, boys overimitate more than girls when two tools are used during early childhood but this effect disappears with age, whereas girls overimitate more than boys when one tool is used in late childhood, although the interaction between sex of the child and tool number was not significant in the present study. All in all, our results offer exciting new avenues for future research regarding sex differences during childhood in social learning.

To conclude, we showed that overimitation is influenced by instrumental factors such as the number of tools used in the demonstration, with children overimitating more when one tool is used to perform both relevant and irrelevant actions than when two tools are used to perform these actions. The task history also affected overimitation as being previously unsuccessful in the task generated more overimitative behaviour afterwards. Finally, children's sex also predicted their tendency to overimitate, with girls overimitating more than boys at age 9–10 years. However, we did not find a significant effect of the model-displayed emotions on overimitation, although this may have resulted from our experimental protocol. Moreover, it should be noted that the end-state of the relevant action (i.e. retrieving the bucket with the hook) was not shown during the demonstration and this point deserves further research regarding to what extent it affects overimitation rates. Overall results in this study tend to support the causal account rather than the social account for explaining overimitation; however, we believe that explanations for overimitation depend on many factors which should be considered together accounting for its occurrence and evolutionary relevance.

Ethics. Ethical approvals were received from the University of Edinburgh, Edinburgh, Scotland and the Max Planck Institute of Evolutionary Anthropology, Leipzig, Germany, as well as from the participating schools.

Data accessibility. The data supporting the findings of this study as well as the videos of the demonstrators used in this study are available on the Open Science Framework repository at https://osf.io/bstcz/.

Authors' contributions. A.F. and T.G. conceived the study. A.F. and H.S. collected the data in France and Germany, respectively. H.S. and L.P.S. analysed the data. A.F. wrote the first version of this article and all authors provided critical reviews and approved the final version.

Competing interests. We have no competing interests.

Funding. A.F. was supported by a doctoral scholarship from Suor Orsola Benincasa University and a Research Support Grant from the University of Edinburgh. H.S. was supported by a fellowship of the International Max Planck Research School on Neuroscience of Communication (IMPRS NeuroCom). T.G. was supported by the Swiss National Science foundation (grant nos. CR13I1_162720/1 and PCEFP1_186832).

Acknowledgements. We thank all participating schools that took part in this study. A.F. thanks Helen Wright for helpful comments made on previous versions of this article, Diane Austry for providing help with the double coding and Hélène Ferrandez and William Chevalier for serving as demonstrators in the video recordings.

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
