## [Peer Review File · Royal Society Open Science]

Review History

RSOS-201373.R0 (Original submission)

Review form: Reviewer 1

Is the manuscript scientifically sound in its present form?

No

Are the interpretations and conclusions justified by the results?

Yes

Is the language acceptable?

Yes

Do you have any ethical concerns with this paper?

Yes

Have you any concerns about statistical analyses in this paper?

No

Recommendation?

Reject

Comments to the Author(s)

This manuscript details a study which sought to explore the factors which influence overimitation. First, the authors examined whether irrelevant actions were completed with the same tool needed to complete the task, or a second tool. Second, they examined whether prior success with the object led to reduced overimitation. Third, they examined whether the emotion of the demonstrator predicted overimitation. Finally they looked for sex differences in overimitation.

On the whole, I think some of the ideas for this study were interesting but the authors have tried to examine too many things in one study, leaving it unfocussed and confounded.

My specific comments on the manuscript appear below:

Introduction:

The theoretical grounding for the study could be presented with more nuance. While there are a social cluster of theories of overimitation, these don't necessarily all boil down to social motivation. Indeed, normative accounts often don't specify social motivation as a driving force behind overimitation at all. The second paragraph on page three provides a very simplistic theoretical account which is not entirely accurate.

The social motivation account is labelled as a 'gold standard' but I think this is far from accepted by researchers in this field.

It could be useful for the authors to tie their hypotheses in to the theoretical background more clearly. It seems that the manipulations that have been chosen in this study don't directly tap in to causal vs social theoretical accounts because this is too simplistic.

Method

The overimitation actions were different for the 1 vs 2 tool comparison. How can the authors be sure that these actions were equally salient, memorable, and difficult? Any number of confounds could drive the difference of overimitation between these conditions so it is difficult to draw any conclusions from any differences observed.

I find the lack of modelled interaction effects a little worrying. Would the authors not predict that prior success could alter the effects of the other manipulations?

Review form: Reviewer 2**Is the manuscript scientifically sound in its present form?**

Yes

Are the interpretations and conclusions justified by the results?

Yes

Is the language acceptable?

Yes

Do you have any ethical concerns with this paper?

No

Have you any concerns about statistical analyses in this paper?

No

Recommendation?

Accept with minor revision (please list in comments)

Comments to the Author(s)

This manuscript describes a study that investigates over-imitation in a tool-based task, the hook task with 9-10-year-old French and German children. First, the authors established the children's baseline success and unsucess on the task before showing them a video to determine how the number of tools used by the demonstrator (one or two), and the demonstrator's emotional state (neutral or positive) impacted the child's subsequent performance with the task when presented with it for a second time. As previous research has shown some sex differences in children's likelihood to over-imitate the authors also included the participant's sex in their analysis. The question investigated by this study and the use of multiple tools in the demonstration is interesting and important. I thought this manuscript was concise, well written, and an interesting new avenue for over-imitation research. Notwithstanding, I have a few queries and comments that I have laid out below:

I thought the methodological design of the study having a pre-demonstration phase to get a measure of the children's baseline success with the apparatus, then a demonstration of the irrelevant and relevant actions, and post-demonstration phase for the children to potentially copy was well-designed. I did have a couple of questions regarding the models in the video. Were these male and female demonstrators familiar to the child participants? Were they from the same localities as where the children were tested?

On page 2, the authors point to chimpanzees as the prolific non-human tool users but they do not discuss birds who are also avid tool users. Some species of birds such as New Caledonian Crows have been shown to manipulate tools away from the sticks, sponges, and hammers of chimpanzees (e.g. Hunt, 1996; Chappell & Kacelnik, 2002). Perhaps the authors could include a brief discussion of other successful tool-using animals?

In the participant section of the methods (page 4), the authors note on lines 47-48 that SES and race information was not systemically collected but they state that participants were from middle to high backgrounds and mostly Caucasian. Where did the authors ascertain this information from if it was not assessed?

In the coding section, the authors mentioned that only those children that made a hook from the pipe cleaner were classified as successful. Were any children successful using other techniques (such as dragging the basket out of the bottle with the pipe cleaner) rather than making a hook? OR where these children also classified as unsuccessful?

In the results section although the authors point to counterbalancing the sex of the demonstrator they do not include any analysis to say whether the sex of the demonstrator had any impact on the participant's subsequent over-imitation results. I wondered if this analysis was also done? I also had some follow questions based on how the participant's history (pre-demonstration over-imitation success) impacted their post-demonstration over-imitation success that I wasn't clear on. Although task history impacted children's likelihood to over-imitate, how was this likelihood tied to the participant's use of one or two tools previously, and their subsequent tool use after the demonstration? Basically, how were the participant's unsucess and success related to the

number of tools they used? Also, did any of the previously successful children replicate any of the irrelevant actions after the demonstration or did they all maintain their initial behaviour?

On page 7 lines 30-33, the authors suggest some potential sex and the number of tools used interactions, and I wondered if the authors could include a number of tools by sex interaction in their model to see if they get a similar result in their sample?

Some minor queries/comments:

On page 3, line 10-13 I think it would be more appropriate to describe these countries as Non-WEIRD (according to Henrich, Heine, & Norenzayan, 2010) rather than non-Western cultures.

On page 4, lines 49-54 the authors provide model design details, perhaps, this information may be better placed in the data analysis sub-section rather than participants section?

In the materials and procedures section, the authors mention that the experimenters were French and German nationals. What was the procedure for translating and back-translating the instructions from English to French and German, respectively for reliability across locations?

The experimental protocol offered children up to a minute to retrieve the reward from inside the basket. Although the majority of children retrieve the reward within the first minute, some studies have used up to 3 minutes (e.g. see Rawlings, 2018) to extend the window of time for success. Was there a reason why the authors used 1 minute?

On page 7, line 3, the authors suggest an alternative sentence, "This is funny what I am doing" as a potential example to elicit a positive attitude towards an experimenter. Would the authors suggest using humour in this manner would be more effective at eliciting a positive attitude than affirming sentences such as, "I'm doing a good job?"

On page 7, line 44, I believe that "may" is missing in this sentence? So, the sentence would now read, "although this may have resulted"

I wasn't sure what the different shades of grey illustrate in Figure 1 as I didn't see these mentioned in the figure caption?

Do the authors have a copy of the video file used in the demonstration that they could include in their submission? I was interested to see if the loop made at the top of the pipe cleaner could have been thought of as a 'handle' to use the pipe cleaner rather than as a completely irrelevant action?

The details for reference 57 appear to be incomplete?

I believe that reference 14 and 52 are referring to the same paper but the citation is slightly different?

In the supplementary materials, the authors use gender and male/female but in the article, they use sex and boy/girl.

I have also included some suggestions of potential other literature for discussion in the manuscript. In the summary section (lines 28-19) the authors refer to most over-imitation research being conducted with task naïve participants with a single tool, perhaps they could include the alternative over-imitation research without tools in the text e.g. imitative fidelity studies using necklace making tasks by Clegg and Legare, 2015; 2016? On page 3, lines 38-40, to further substantiate their claim that over-imitation still persists when social factors are removed in

experimental settings, the authors could also include a discussion of the study done by Whiten et al, 2016 with similar evidence from children and adults in naturalistic contexts. A recent over-imitation book chapter by Rawlings et al, 2019 could also be included in the references.

Decision letter (RSOS-201373.R0)

Dear Dr Frick

The Editors assigned to your paper RSOS-201373 "Carry-over effects of tool functionality and previous unsuccessfulness increase overimitation in children" have now received comments from reviewers and would like you to revise the paper in accordance with the reviewer comments and any comments from the Editors. Please note this decision does not guarantee eventual acceptance.

Please submit your revised manuscript and required files (see below) no later than 21 days from today's (ie 05-Nov-2020) date. Note: the ScholarOne system will 'lock' if submission of the revision is attempted 21 or more days after the deadline. If you do not think you will be able to meet this deadline please contact the editorial office immediately.

on behalf of Dr Teodora Gliga (Associate Editor) and Essi Viding (Subject Editor)

Associate Editor Comments to Author (Dr Teodora Gliga):

Comments to the Author:

Please revise your manuscript in accord with the reviewer's comments, paying particular attention to expanding and clarifying the theoretical background as well as providing missing methodological detail. Both reviewers noted that lack of detail and clarity in your methods means that potential confounds make your findings uninterpretable.

Reviewer comments to Author:

Reviewer: 1

Comments to the Author(s)

This manuscript details a study which sought to explore the factors which influence overimitation. First, the authors examined whether irrelevant actions were completed with the same tool needed to complete the task, or a second tool. Second, they examined whether prior success with the object led to reduced overimitation. Third, they examined whether the emotion of the demonstrator predicted overimitation. Finally they looked for sex differences in overimitation.

On the whole, I think some of the ideas for this study were interesting but the authors have tried to examine too many things in one study, leaving it unfocussed and confounded.

My specific comments on the manuscript appear below:

Introduction:

The theoretical grounding for the study could be presented with more nuance. While there are a social cluster of theories of overimitation, these don't necessarily all boil down to social motivation. Indeed, normative accounts often don't specify social motivation as a driving force behind overimitation at all. The second paragraph on page three provides a very simplistic theoretical account which is not entirely accurate.

The social motivation account is labelled as a 'gold standard' but I think this is far from accepted by researchers in this field.

It could be useful for the authors to tie their hypotheses in to the theoretical background more clearly. It seems that the manipulations that have been chosen in this study don't directly tap in to causal vs social theoretical accounts because this is too simplistic.

Method

The overimitation actions were different for the 1 vs 2 tool comparison. How can the authors be sure that these actions were equally salient, memorable, and difficult? Any number of confounds could drive the difference of overimitation between these conditions so it is difficult to draw any conclusions from any differences observed.

I find the lack of modelled interaction effects a little worrying. Would the authors not predict that prior success could alter the effects of the other manipulations?

Reviewer: 2

Comments to the Author(s)

This manuscript describes a study that investigates over-imitation in a tool-based task, the hook task with 9-10-year-old French and German children. First, the authors established the children's baseline success and unsuccess on the task before showing them a video to determine how the number of tools used by the demonstrator (one or two), and the demonstrator's emotional state (neutral or positive) impacted the child's subsequent performance with the task when presented with it for a second time. As previous research has shown some sex differences in children's likelihood to over-imitate the authors also included the participant's sex in their analysis. The question investigated by this study and the use of multiple tools in the demonstration is interesting and important. I thought this manuscript was concise, well written, and an interesting new avenue for over-imitation research. Notwithstanding, I have a few queries and comments that I have laid out below:

I thought the methodological design of the study having a pre-demonstration phase to get a measure of the children's baseline success with the apparatus, then a demonstration of the irrelevant and relevant actions, and post-demonstration phase for the children to potentially copy was well-designed. I did have a couple of questions regarding the models in the video. Were these male and female demonstrators familiar to the child participants? Were they from the same localities as where the children were tested?

On page 2, the authors point to chimpanzees as the prolific non-human tool users but they do not discuss birds who are also avid tool users. Some species of birds such as New Caledonian Crows have been shown to manipulate tools away from the sticks, sponges, and hammers of chimpanzees (e.g. Hunt, 1996; Chappell & Kacelnik, 2002). Perhaps the authors could include a brief discussion of other successful tool-using animals?

In the participant section of the methods (page 4), the authors note on lines 47-48 that SES and race information was not systemically collected but they state that participants were from middle to high backgrounds and mostly Caucasian. Where did the authors ascertain this information from if it was not assessed?

In the coding section, the authors mentioned that only those children that made a hook from the pipe cleaner were classified as successful. Were any children successful using other techniques (such as dragging the basket out of the bottle with the pipe cleaner) rather than making a hook? OR where these children also classified as unsuccessful?

In the results section although the authors point to counterbalancing the sex of the demonstrator they do not include any analysis to say whether the sex of the demonstrator had any impact on the participant's subsequent over-imitation results. I wondered if this analysis was also done? I also had some follow questions based on how the participant's history (pre-demonstration over-imitation success) impacted their post-demonstration over-imitation success that I wasn't clear on. Although task history impacted children's likelihood to over-imitate, how was this likelihood tied to the participant's use of one or two tools previously, and their subsequent tool use after the demonstration? Basically, how were the participant's unsuccess and success related to the number of tools they used? Also, did any of the previously successful children replicate any of the irrelevant actions after the demonstration or did they all maintain their initial behaviour?

On page 7 lines 30-33, the authors suggest some potential sex and the number of tools used interactions, and I wondered if the authors could include a number of tools by sex interaction in their model to see if they get a similar result in their sample?

Some minor queries/comments:

On page 3, line 10-13 I think it would be more appropriate to describe these countries as Non-WEIRD (according to Henrich, Heine, & Norenzayan, 2010) rather than non-Western cultures.

On page 4, lines 49-54 the authors provide model design details, perhaps, this information may be better placed in the data analysis sub-section rather than participants section?

In the materials and procedures section, the authors mention that the experimenters were French and German nationals. What was the procedure for translating and back-translating the instructions from English to French and German, respectively for reliability across locations?

The experimental protocol offered children up to a minute to retrieve the reward from inside the basket. Although the majority of children retrieve the reward within the first minute, some studies have used up to 3 minutes (e.g. see Rawlings, 2018) to extend the window of time for success. Was there a reason why the authors used 1 minute?

On page 7, line 3, the authors suggest an alternative sentence, "This is funny what I am doing" as a potential example to elicit a positive attitude towards an experimenter. Would the authors suggest using humour in this manner would be more effective at eliciting a positive attitude than affirming sentences such as, "I'm doing a good job?"

On page 7, line 44, I believe that "may" is missing in this sentence? So, the sentence would now read, "although this may have resulted"

I wasn't sure what the different shades of grey illustrate in Figure 1 as I didn't see these mentioned in the figure caption?

Do the authors have a copy of the video file used in the demonstration that they could include in their submission? I was interested to see if the loop made at the top of the pipe cleaner could have been thought of as a 'handle' to use the pipe cleaner rather than as a completely irrelevant action?

The details for reference 57 appear to be incomplete?

I believe that reference 14 and 52 are referring to the same paper but the citation is slightly different?

In the supplementary materials, the authors use gender and male/female but in the article, they use sex and boy/girl.

I have also included some suggestions of potential other literature for discussion in the manuscript. In the summary section (lines 28-19) the authors refer to most over-imitation research being conducted with task naïve participants with a single tool, perhaps they could include the alternative over-imitation research without tools in the text e.g. imitative fidelity studies using necklace making tasks by Clegg and Legare, 2015; 2016? On page 3, lines 38-40, to further substantiate their claim that over-imitation still persists when social factors are removed in experimental settings, the authors could also include a discussion of the study done by Whiten et al, 2016 with similar evidence from children and adults in naturalistic contexts. A recent over-imitation book chapter by Rawlings et al, 2019 could also be included in the references.

===PREPARING YOUR MANUSCRIPT===

===PREPARING YOUR REVISION IN SCHOLARONE===

- An editable file of each table (.doc, .docx, .xls, .xlsx, or .csv).
- An editable file of all figure and table captions.

- Any electronic supplementary material (ESM).
- If you are requesting a discretionary waiver for the article processing charge, the waiver form must be included at this step.
- If you are providing image files for potential cover images, please upload these at this step, and inform the editorial office you have done so. You must hold the copyright to any image provided.
- A copy of your point-by-point response to referees and Editors. This will expedite the preparation of your proof.

- Ensure that your data access statement meets the requirements at <https://royalsociety.org/journals/authors/author-guidelines/#data>. You should ensure that you cite the dataset in your reference list. If you have deposited data etc in the Dryad repository, please include both the 'For publication' link and 'For review' link at this stage.
- If you are requesting an article processing charge waiver, you must select the relevant waiver option (if requesting a discretionary waiver, the form should have been uploaded at Step 3 'File upload' above).
- If you have uploaded ESM files, please ensure you follow the guidance at <https://royalsociety.org/journals/authors/author-guidelines/#supplementary-material> to include a suitable title and informative caption. An example of appropriate titling and captioning may be found at [https://figshare.com/articles/Table_S2_from_Is_there_a_trade-off_between_peak_performance_and_performance_breadth_across_temperatures_for_aerobic_sc](https://figshare.com/articles/Table_S2_from_Is_there_a_trade-off_between_peak_performance_and_performance_breadth_across_temperatures_for_aerobic_scope_in_teleost_fishes_/3843624) ope_in_teleost_fishes_/3843624.

Author's Response to Decision Letter for (RSOS-201373.R0)

See Appendix A.

RSOS-201373.R1 (Revision)

Review form: Reviewer 2

Is the manuscript scientifically sound in its present form?

Yes

Are the interpretations and conclusions justified by the results?

Yes

Is the language acceptable?

Yes

Do you have any ethical concerns with this paper?

No

Have you any concerns about statistical analyses in this paper?

No

Recommendation?

Accept with minor revision (please list in comments)

Comments to the Author(s)

Thank you to the authors for addressing the previous comments. I think the manuscript is improved but have a few minor points that I have raised below.

Thank you for uploading the demonstration videos. After seeing the video, I think it is important for the authors to note that although the demonstrator makes the irrelevant and relevant actions, they do not retrieve the reward from the plastic tubing during the demonstration. Perhaps the authors would have found different over-imitation rates had they included the hook action in use at the end of the demonstration?

Additionally, I would add that the children were recruited from French and German locations into the abstract (line 31) to clarify the population sample.

Perhaps it may be beneficial to include a discussion of the theoretical rationale to the study in the abstract and the relevance to the developmental research (line 41-42)?

There is a typo on Line 192 where an 's' is missing from "action"

Line 410-414, the authors suggest an age-sex-number of tools effect may be evident but did not include any age analyses in their sample. Was the sample of 9 and 10-year old's too small to examine any relationships?

The references have some outstanding formatting issues:

- Refs on lines 447 and 449 are both for the Clay and Tennie study but with two different years.
- Please check the formatting for the following references on lines 478, 489, 497, 513, 519, 545, 549, 555, 559, 562, 565, 567, 570, 573, 622, 632.

Ahead of table S1 in the supplementary materials and in the post demonstration phase description, the authors appear to have an unfinished sentence, "using both tools: either inserting the string into the bottle" What is the other action referred to with this sentence?

I'm confused by the percentage of success and unsuccessful in Tables S1 and S2. I'm unclear by the number of successful and unsuccessful children using either the string, pipe cleaner, or both during the test sessions with either the one tool or two tool conditions. The percentages of successful and unsuccessful children in each condition do not appear to reach 100%. Also, where is the number of children who used the pipe cleaner in the two tools condition? The subtotal indicates that children did use the string, but overall 0% of children are shown as being unsuccessful or successful with the string.

Table S4 contains both child's gender and child's sex. I believe these are referring to the same variable?

Decision letter (RSOS-201373.R1)

Dear Dr Frick

On behalf of the Editors, we are pleased to inform you that your Manuscript RSOS-201373.R1 "Carry-over effects of tool functionality and previous unsuccessfulness increase overimitation in children" has been accepted for publication in Royal Society Open Science subject to minor revision in accordance with the referees' reports. Please find the referees' comments along with any feedback from the Editors below my signature.

Please submit your revised manuscript and required files (see below) no later than 7 days from today's (ie 28-Apr-2021) date. Note: the ScholarOne system will 'lock' if submission of the revision is attempted 7 or more days after the deadline. If you do not think you will be able to meet this deadline please contact the editorial office immediately.

Kind regards,
Royal Society Open Science Editorial Office

on behalf of Dr Teodora Gliga (Associate Editor) and Essi Viding (Subject Editor)

Associate Editor Comments to Author (Dr Teodora Gliga):

Associate Editor: 1

Comments to the Author:

Thank you for returning your revised manuscript. This revised version was now seen by one of the original reviewers who is pleased with the additional clarifications brought. I read the manuscript myself and I believe that it will make an important contribution to our understanding

of overimitation. While I advise for this paper to be accepted, you will have to provide a new version in which you take into account the remaining comments made by this Reviewer. In particular, the manuscript requires further clarifications - i know that tables S1 and S2 were requested by one of the reviewers yet it remains unclear whether they are adding any clarifications or ruling out confounds pertaining to your main analysis. So either clearly explain this relevance in the main text or remove these tables. Please also carefully check the formatting of references and proofread the text.

Reviewer comments to Author:

Reviewer: 2

Comments to the Author(s)

Thank you to the authors for addressing the previous comments. I think the manuscript is improved but have a few minor points that I have raised below.

Thank you for uploading the demonstration videos. After seeing the video, I think it is important for the authors to note that although the demonstrator makes the irrelevant and relevant actions, they do not retrieve the reward from the plastic tubing during the demonstration. Perhaps the authors would have found different over-imitation rates had they included the hook action in use at the end of the demonstration?

Additionally, I would add that the children were recruited from French and German locations into the abstract (line 31) to clarify the population sample.

Perhaps it may be beneficial to include a discussion of the theoretical rationale to the study in the abstract and the relevance to the developmental research (line 41-42)?

There is a typo on Line 192 where an 's' is missing from "action"

Line 410-414, the authors suggest an age-sex-number of tools effect may be evident but did not include any age analyses in their sample. Was the sample of 9 and 10-year old's too small to examine any relationships?

The references have some outstanding formatting issues:

-Refs on lines 447 and 449 are both for the Clay and Tennie study but with two different years.

-Please check the formatting for the following references on lines 478, 489, 497, 513, 519, 545, 549, 555, 559, 562, 565, 567, 570, 573, 622, 632.

Ahead of table S1 in the supplementary materials and in the post demonstration phase description, the authors appear to have an unfinished sentence, "using both tools: either inserting the string into the bottle" What is the other action referred to with this sentence?

I'm confused by the percentage of success and unsuccessful in Tables S1 and S2. I'm unclear by the number of successful and unsuccessful children using either the string, pipe cleaner, or both during the test sessions with either the one tool or two tool conditions. The percentages of successful and unsuccessful children in each condition do not appear to reach 100%. Also, where is the number of children who used the pipe cleaner in the two tools condition? The subtotal indicates that children did use the string, but overall 0% of children are shown as being unsuccessful or successful with the string.

Table S4 contains both child's gender and child's sex. I believe these are referring to the same variable?

===PREPARING YOUR MANUSCRIPT===

===PREPARING YOUR REVISION IN SCHOLARONE===

-- Ensure that your data access statement meets the requirements at <https://royalsociety.org/journals/authors/author-guidelines/#data>. You should ensure that you cite the dataset in your reference list. If you have deposited data etc in the Dryad repository, please only include the 'For publication' link at this stage. You should remove the 'For review' link.

-- If you have uploaded ESM files, please ensure you follow the guidance at <https://royalsociety.org/journals/authors/author-guidelines/#supplementary-material> to include a suitable title and informative caption. An example of appropriate titling and captioning may be found at https://figshare.com/articles/Table_S2_from_Is_there_a_trade-off_between_peak_performance_and_performance_breadth_across_temperatures_for_aerobic_sc_ope_in_teleost_fishes_/3843624.

Author's Response to Decision Letter for (RSOS-201373.R1)

See Appendix B.

Decision letter (RSOS-201373.R2)

Dear Dr Frick,

I am pleased to inform you that your manuscript entitled "Carry-over effects of tool functionality and previous unsuccessfulness increase overimitation in children" is now accepted for publication in Royal Society Open Science.

You can expect to receive a proof of your article in the near future. Please contact the editorial office (openscience@royalsociety.org) and the production office (openscience_proofs@royalsociety.org) to let us know if you are likely to be away from e-mail contact – if you are going to be away, please nominate a co-author (if available) to manage the proofing process, and ensure they are copied into your email to the journal. Due to rapid publication and an extremely tight schedule, if comments are not received, your paper may experience a delay in publication.

on behalf of Dr Teodora Gliga (Associate Editor) and Essi Viding (Subject Editor)
openscience@royalsociety.org

Follow Royal Society Publishing on Twitter: @RSocPublishing
Follow Royal Society Publishing on Facebook:
<https://www.facebook.com/RoyalSocietyPublishing.FanPage/>

Read Royal Society Publishing's blog:
<https://royalsociety.org/blog/blogsearchpage/?category=Publishing>

Appendix A

Associate Editor Comments to Author (Dr Teodora Gliga):

Comments to the Author:

Please revise your manuscript in accord with the reviewer's comments, paying particular attention to expanding and clarifying the theoretical background as well as providing missing methodological detail. Both reviewers noted that lack of detail and clarity in your methods means that potential confounds make your findings uninterpretable.

We thank you very much for your review of our manuscript "**Carry-over effects of tool functionality and previous unsuccessfulness increase overimitation in children**". We believe we have now addressed the two reviewers' comments, particularly addressing the potential problem of confounds raised by Reviewer 1. We believe our findings are now easier to interpret, and our manuscript clearer thanks to the comments of the reviewers and we describe in the following how we addressed their comments.

Reviewer 1

- 1. On the whole, I think some of the ideas for this study were interesting but the authors have tried to examine too many things in one study, leaving it unfocussed and confounded.**

We thank Reviewer 1 for their comment on the interest of our paper. We are sorry that Reviewer 1 finds our study unfocussed and confounded. We acknowledge that our paper potentially looks at too many variables in one study. However, as mentioned by Reviewer 2, these questions are timely important, and our results have a potential theoretical and practical impact for future studies in the field of social learning, tool-use, affective science and sex differences.

We would like to kindly recall that the present paper was submitted as a follow-up to a previous paper published in the same journal (Frick et al. 2017). In this paper, two ideas (the effects of previous unsuccessfulness and number of tools) were left answered as noted by previous Reviewers, and the present paper aimed to address these two questions. Finally, there was a notable sex effect, subsequently reproduced by several studies, that was worth investigating. We also added a simple manipulation to investigate the potential effect of the emotion displayed by the demonstrator as no previous study has explored this question and we wanted to provide first results regarding this question. As a consequence, we acknowledge that the present paper addresses several important unanswered questions in one space, although we do not consider their breadth to confound our conclusions, which we have attempted to streamline in our paper.

Moreover, based on the comments of both Reviewer 1 and 2, we now provide an investigation of all the two-way interactions in a statistical model to clarify whether the result for one factor is

not confounded by another factor. As all two-way interactions are non-significant (see Supplementary Material), we are now confident that our results are not confounded.

- 2. The theoretical grounding for the study could be presented with more nuance. While there are a social cluster of theories of overimitation, these don't necessarily all boil down to social motivation. Indeed, normative accounts often don't specify social motivation as a driving force behind overimitation at all. The second paragraph on page three provides a very simplistic theoretical account which is not entirely accurate.**

We thank Reviewer 1 for that note. We now provide a more nuanced description of these two accounts, specifying that social affiliative and normative accounts are different from each other (L90-L91). However, given that the normative accounts focus on children perceiving irrelevant actions as social norms, we believe that they fall under the umbrella term of social motivations because if children overimitate in this context it is because they are socially motivated to follow the norms (L91-L195).

- 3. The social motivation account is labelled as a 'gold standard' but I think this is far from accepted by researchers in this field.**

We agree with Reviewer 1 that the term 'gold standard' is an over-statement and we have therefore deleted this term from the manuscript.

- 4. It could be useful for the authors to tie their hypotheses in to the theoretical background more clearly. It seems that the manipulations that have been chosen in this study don't directly tap in to causal vs social theoretical accounts because this is too simplistic.**

We thank the reviewer for pointing out the need to clarify the rationale of our study according to the causal and social theoretical accounts. In the Introduction, we explain more precisely how each investigated factor tap onto one or another account (tool functionality and previous successfulness for the causal account and emotion displayed by the demonstrator for the social account; L123-L125). Moreover, before describing the literature regarding each factor, we now make clear which account they relate to at the beginning of each paragraph (L126-L128, L144-L145 and L168-L170).

- 5. The overimitation actions were different for the 1 vs 2 tool comparison. How can the authors be sure that these actions were equally salient, memorable, and difficult? Any number of confounds could drive the difference of overimitation between these conditions so it is difficult to draw any conclusions from any differences observed.**

We thank Reviewer 1 for this comment and the opportunity to develop on the differences between the one tool and two tools condition. We are confident that these actions were similarly difficult because each action consisted in one action, that is making a loop or a circle. Although making a loop taps into finer motor abilities, children in the targeted age group in our study were old enough to perform this action without any difficulties. Whether or not they were both similarly memorable and salient was actually the aim of this comparison. As discussed in our Discussion (L330-L341), it is very likely that the difference between these actions trigger different processes and were therefore different in nature given how they would be perceived by the children. This is an important finding that children overimitated more in the one tool condition than in the two-tool condition because it has practical implications for future overimitation research regarding which actions to overimitate the researchers choose, but also theoretical implications because it provides the first evidence that different types of irrelevant actions trigger different probabilities of overimitation.

6. I find the lack of modelled interaction effects a little worrying. Would the authors not predict that prior success could alter the effects of the other manipulations?

We did not draw any predictions regarding an effect of prior success on the other manipulation. However, we agree with Reviewer 1 that this point should be further examined. In fact, as stated previously, we now provide a statistical model looking at all the two-way interactions, which allows for the investigation of whether or not prior success had an effect on other factor (see Supplementary Material). None of these interactions provided sufficient evidence to support an interaction hypothesis that readers might have.

Reviewer: 2

We thank Reviewer 2 very much for the general comment and summary on our manuscript and we are happy that they find that our study taps important questions related to social learning and tool-use, as well as sex differences. We are happy to address all the points raised by Reviewer 2 in what follows.

1. I did have a couple of questions regarding the models in the video. Were these male and female demonstrators familiar to the child participants? Were they from the same localities as where the children were tested?

This is indeed an important point to clarify. Both demonstrators were unfamiliar to the children and were from a different locality than where the children were tested. We now clarify this point in the Method section (L227-L228).

- 2. On page 2, the authors point to chimpanzees as the prolific non-human tool users but they do not discuss birds who are also avid tool users. Some species of birds such as New Caledonian Crows have been shown to manipulate tools away from the sticks, sponges, and hammers of chimpanzees (e.g. Hunt, 1996; Chappell & Kacelnik, 2002). Perhaps the authors could include a brief discussion of other successful tool-using animals?**

We thank Reviewer 2 for this suggestion, which is totally fair. We now include the example of New Caledonian Crows that have been reported to use and even make new tools from raw materials both in the wild and in captivity (L51-L55).

- 3. In the participant section of the methods (page 4), the authors note on lines 47-48 that SES and race information was not systematically collected but they state that participants were from middle to high backgrounds and mostly Caucasian. Where did the authors ascertain this information from if it was not assessed?**

We now provide a precise percentage of the children who were Caucasian based on the inspection of the video recordings. For the SES, we base our estimation given the area where the data was collected (L199-L201).

- 4. In the coding section, the authors mentioned that only those children that made a hook from the pipe cleaner were classified as successful. Were any children successful using other techniques (such as dragging the basket out of the bottle with the pipe cleaner) rather than making a hook? OR were these children also classified as unsuccessful?**

We thank Reviewer 2 for pointing out the need to clarify this point. After reviewing all the videos, we noted that except one child, all successful children made a hook to retrieve the bucket from the bottle during the pre-demonstration phase, and all successful children made a hook in the post-demonstration phase. We have now changed our coding to include the child who did not make a hook by classifying successful children if they retrieved the reward, this avoided to exclude a child from a sample (L244-L247). We also now add the information that one child did not make a hook to solve the Hook task in the Results section (L294-L297).

- 5. In the results section although the authors point to counterbalancing the gender of the demonstrator they do not include any analysis to say whether the gender of the demonstrator had any impact on the participant's subsequent over-imitation results. I wondered if this analysis was also done?**

We thank Reviewer 2 for this idea as we did not include this in the analyses. We now add a pre-analysis, in which we investigated whether children's or demonstrator's sex had an effect on children's over-imitation tendencies. Since only children's sex seemed to have a significant effect, we included that variable in our main analysis and omitted demonstrator's sex in this analysis (see Supplementary Material for details and in L264-L268).

6. I also had some follow questions based on how the participant's history (pre-demonstration over-imitation success) impacted their post-demonstration over-imitation success that I wasn't clear on.

We now add a model in our Methods section and Supplementary Material and Results section that specifically looks at the effect of prior-success on post-success, revealing no significant effect (L261-L263, L302-L303 and Supplementary Material).

7. Although task history impacted children's likelihood to over-imitate, how was this likelihood tied to the participant's use of one or two tools previously, and their subsequent tool use after the demonstration? Basically, how were the participant's success and failure related to the number of tools they used?

We thank Reviewer 2 for this interesting note, which should be of interest for researchers working on innovation and tool-use in children. We now provide this information in Table S1 and S2 in the Supplemental Material.

8. Also, did any of the previously successful children replicate any of the irrelevant actions after the demonstration or did they all maintain their initial behaviour?

This result is indicated in our Results section in which we report that 51% of the successful children did reproduce the irrelevant action whereas 49% maintained their initial behaviour (L313).

9. On page 7 lines 30-33, the authors suggest some potential gender and the number of tools used interactions, and I wondered if the authors could include a number of tools by gender interaction in their model to see if they get a similar result in their sample?

We thank Reviewer 2 for this suggestion. We now include all possible two-way interactions in our model. None of these were statistically significant (L287-L289). As such, we now discuss that a potential interaction between sex and tool number could be an interesting venue for future research, although the data from our sample does not seem to support this idea (L410-L414).

10. On page 3, line 10-13 I think it would be more appropriate to describe these countries as Non-WEIRD (according to Henrich, Heine, & Norenzayan, 2010) rather than non-Western cultures.

We agree with Reviewer 2 and we now describe these countries according to the term used by Henrich et al. (2010) in L77-L82.

11. On page 4, lines 49-54 the authors provide model design details, perhaps, this information may be better placed in the data analysis sub-section rather than participants section?

We agree with Reviewer 2 and we now provide this information in the Data Analyses sub-section instead of the Participants sub-section (L269-L276).

12. In the materials and procedures section, the authors mention that the experimenters were French and German nationals. What was the procedure for translating and back-translating the instructions from English to French and German, respectively for reliability across locations?

The instructions were initially written in English and then translated in French and German by the two experimenters, who are residing in English-speaking countries and are therefore fluent in English (L209-L210).

13. The experimental protocol offered children up to a minute to retrieve the reward from inside the basket. Although the majority of children retrieve the reward within the first minute, some studies have used up to 3 minutes (e.g. see Rawlings, 2018) to extend the window of time for success. Was there a reason why the authors used 1 minute?

We used one minute as a time limit to ensure comparisons with other studies using the same task with children (Beck et al., 2011; Frick et al., 2017). This is now explicitly stated in L221-L223.

14. On page 7, line 3, the authors suggest an alternative sentence, “This is funny what I am doing” as a potential example to elicit a positive attitude towards an experimenter. Would the authors suggest using humour in this manner would be more effective at eliciting a positive attitude than affirming sentences such as, “I’m doing a good job?”

We thank Reviewer 2 for raising this point. We think that a playful attitude would elicit more positive attitude and potentially increase overimitation (L380).

15. On page 7, line 44, I believe that “may” is missing in this sentence? So, the sentence would now read, “although this may have resulted”

We thank Reviewer 2 for this, and it was indeed a missing word that we added (L422).

16. I wasn't sure what the different shades of grey illustrate in Figure 1 as I didn't see these mentioned in the figure caption?

We are sorry for the lack of clarity. The darker the circles, the more it indicates that children overimitated. We now mention this detail in the caption (L677-679).

17. Do the authors have a copy of the video file used in the demonstration that they could include in their submission? I was interested to see if the loop made at the top of the pipe cleaner could have been thought of as a ‘handle’ to use the pipe cleaner rather than as a completely irrelevant action?

We have uploaded the videos of the demonstrators with all the other material (data) for researchers interested in looking at our material. We appreciated the note by Reviewer 2 and now nuanced, or at least discussed the possibility that the children perceived the loop as a handle (L338-341).

18. The details for reference 57 appear to be incomplete?

We thank Reviewer 2 for this detail, we have updated the reference which is now published.

19. I believe that reference 14 and 52 are referring to the same paper but the citation is slightly different?

We thank Reviewer 2 again for this detail, it is indeed the same paper and was an error due to the citation software.

20. In the supplementary materials, the authors use gender and male/female but in the article, they use gender and boy/girl.

We now use the term ‘sex’ alongside with the two terms ‘male’ and ‘female’ for the demonstrators and ‘boys’ and ‘girls’ when referring to the sex of the children.

21. I have also included some suggestions of potential other literature for discussion in the manuscript. In the summary section (lines 28-19) the authors refer to most over-

imitation research being conducted with task naïve participants with a single tool, perhaps they could include the alternative over-imitation research without tools in the text e.g. imitative fidelity studies using necklace making tasks by Clegg and Legare, 2015; 2016? On page 3, lines 38-40, to further substantiate their claim that over-imitation still persists when social factors are removed in experimental settings, the authors could also include a discussion of the study done by Whiten et al, 2016 with similar evidence from children and adults in naturalistic contexts. A recent over-imitation book chapter by Rawlings et al, 2019 could also be included in the references.

These are interesting and fair suggestions. These references have been added in the manuscript (L134-L136, L114-L116 and L66-67, respectively).

Appendix B

Associate Editor Comments to Author (Dr Teodora Gliga):

Associate Editor: 1

Comments to the Author:

Thank you for returning your revised manuscript. This revised version was now seen by one of the original reviewers who is pleased with the additional clarifications brought. I read the manuscript myself and I believe that it will make an important contribution to our understanding of overimitation. While I advise for this paper to be accepted, you will have to provide a new version in which you take into account the remaining comments made by this Reviewer. In particular, the manuscript requires further clarifications - i know that tables S1 and S2 were requested by one of the reviewers yet it remains unclear whether they are adding any clarifications or ruling out confounds pertaining to your main analysis. So either clearly explain this relevance in the main text or remove these tables. Please also carefully check the formatting of references and proofread the text.

Response: We thank you very much for this comment and we are very happy about the decision made on accepting our manuscript for publication in *Royal Society Open Science*. We agree that the tables S1 and S2 do not provide relevant information concerning our main analysis. We have therefore removed these tables, although we have kept in our data file the different variables that were presented in the tables, as such interested readers, and perhaps especially those working on tool-use and tool-innovation, would have the information regarding how the string was used (either only touched or inserted). We have carefully checked the formatting of all the references and proof-read the text by one of the authors who a native English speaker and another native English speaker.

Reviewer comments to Author:

Reviewer: 2

Comments to the Author(s)

Thank you to the authors for addressing the previous comments. I think the manuscript is improved but have a few minor points that I have raised below.

Response: We thank very much the Reviewer for the previous helpful comments on our manuscript and we are happy to address the new minor points raised below.

Thank you for uploading the demonstration videos. After seeing the video, I think it is important for the authors to note that although the demonstrator makes the irrelevant and relevant actions, they do not retrieve the reward from the plastic tubing during the demonstration. Perhaps the authors would have found different over-imitation rates had they included the hook action in use at the end of the demonstration?

Response: This is indeed a point that deserves to be raised and should be tackled in future studies. We highlighted this fact now in the methods section (L423-L425). Additionally, we

now mention at the end of our discussion that comparing demonstrations that show the end-state of the actions and to what extent it affects overimitation rates should be targeted by potential future studies (L644-L646).

Additionally, I would add that the children were recruited from French and German locations into the abstract (line 31) to clarify the population sample.

Response: We agree with the Reviewer and have specified that the children tested in this study were French and German in the abstract (L27).

Perhaps it may be beneficial to include a discussion of the theoretical rationale to the study in the abstract and the relevance to the developmental research (line 41-42)?

Response: We agree with the Reviewer and added the respective information in the abstract (L21-24, L34-35)

There is a typo on Line 192 where an 's' is missing from "action"

Response: The typo has been corrected.

Line 410-414, the authors suggest an age-sex-number of tools effect may be evident but did not include any age analyses in their sample. Was the sample of 9 and 10-year old's too small to examine any relationships?

Response: Yes, the present sample is too restricted in terms of age range to examine age-related differences regarding the number of tools used and sex. We have specified this in our Discussion (L615-L616) where we mention that a future study should be set to test this developmental aspect.

The references have some outstanding formatting issues:

-Refs on lines 447 and 449 are both for the Clay and Tennie study but with two different years.

Response: We have removed the reference Clay and Tennie (2017) which was incorrect.

-Please check the formatting for the following references on lines 478, 489, 497, 513, 519, 545, 549, 555, 559, 562, 565, 567, 570, 573, 622, 632.

Response: We have checked the references on the different lines and corrected the wrong formatting.

Ahead of table S1 in the supplementary materials and in the post demonstration phase description, the authors appear to have an unfinished sentence, "using both tools: either inserting the string into the bottle" What is the other action referred to with this sentence? I'm confused by the percentage of success and unsuccessful in Tables S1 and S2. I'm unclear by the number of successful and unsuccessful children using either the string,

pipe cleaner, or both during the test sessions with either the one tool or two tool conditions. The percentages of successful and unsuccessful children in each condition do not appear to reach 100%? Also, where is the number of children who used the pipe cleaner in the two tools condition? The subtotal indicates that children did use the string, but overall 0% of children are shown as being unsuccessful or successful with the string. Table S4 contains both child's gender and child's sex. I believe these are referring to the same variable?

Response: We are sorry for the lack of clarity. Following the comment from the Associate Editor, we believe that tables S1 and S2 contain information that are not strongly relevant to the main analysis. However, we also believe that this data would be of potential interest for researchers working in the fields of tool-use and tool-innovation. As such, we have deleted the tables from the Supplemental Material but we have kept the coding of the different variables in the data file, which is available online, making it as clear as possible.